# Emotional impact according to the way cancer patients are conducted to the surgical center: A randomized clinical trial comparing ambulation to the stretcher

Gabriela da Silva Oliveira[1,2,3☉¤a*], Paula Moreira da Silva Sabaini[2,3☉],
Priscila Grecca Pedrão[4,5], Vinicius Bars da Silva[1,2,3☉], Marcos Alves de Lima[6,3☉],
Thiago Emilio da Silva Mantovani[3☉], Carlos Eduardo Mattos da Cunha Andrade[1☉],
Ricardo dos Reis[1,3,4,5☉]

1 Department of Gynecological Oncology, Barretos Cancer Hospital, Barretos, São Paulo, Brazil,
2 Researcher Support Center, Barretos Cancer Hospital, Brazil, 3 Research and Teaching Institute,
Barretos Cancer Hospital, Brazil, 4 Molecular Oncology Research Center, Barretos Cancer Hospital,
Barretos, São Paulo, Brasil, 5 HPV Research Group, Barretos Cancer Hospital, Barretos, São Paulo,
Brasil, 6 Division of Epidemiology and Biostatistics, Barretos Cancer Hospital, Brazil

☉ These authors contributed equally to this work.
¤aCurrent Address: Research and Teaching Institute, Barretos Cancer Hospital, 1331 Antenor Duarte Vilela
St, Barretos, São Paulo 14784-400, Brazil
* gabriiela.oliveira@hotmail.com

journal.pone.0320856

University, EGYPT

**Peer Review History:** PLOS recognizes the
benefits of transparency in the peer review
process; therefore, we enable the publication
of all of the content of peer review and
author responses alongside final, published
articles. The editorial history of this article is
available here: https://doi.org/10.1371/journal.
pone.0320856

## Abstract

This study aimed to evaluate anxiety, depression and satisfaction of cancer patients
according to the type of conduction to the surgical center. A randomized clinical trial was
carried out at Barretos Cancer Hospital from 2019 to 2021. One hundred seventy-six
patients were enrolled: 94 in the control group (conduction to the surgical center on a
stretcher and pajamas) and 82 in the experimental group (conduction to the surgical
center by ambulation in one's own clothes). The Hospital Anxiety and Depression Scale
(HADS) was used to assess the levels of anxiety and depression, and the Surgery Health
Care Satisfaction Assessment Questionnaire (Sati-Cir) to assess the patient's satisfaction.
There was no difference regarding anxiety and depression symptoms between groups, 15
(18.3%) and 20 (21.3%) patients had anxiety symptoms in the ambulation group and in the
stretcher group, respectively (p = 0.621). Six (7.3%) patients in the ambulation group and
4 (4.3%) met the criteria for depression (p = 0.518). Compared to those who were referred
to the surgical center on a stretcher, those in the ambulation group were more often
satisfied or very satisfied according to the waiting time until surgery (91.5% vs 74.5%; p =
0.008) and type of clothing, own clothes compared to pajamas (100% vs 79%; p = 0.001).
When all patients were asked about their preferred method of going to the surgical center,
70.5% chose to walk in their own clothes, 28.4% preferred a stretcher, and 1.1% opted for
a wheelchair (p < 0.001). The transportation method to the surgical center did not affect
anxiety and depression levels in cancer patients with no or low fall risk. However, patients
in the ambulation group were more satisfied with the waiting time for surgery and the type
of clothing. Clinical trial registration: clinicaltrials.gov, NCT03576482, https://clinicaltrials.
gov/ct2/show/NCT03576482

**Data availability statement:** The data underlying the results presented in the study are available from: https://doi.org/10.6084/m9.figshare.27926655.v1

**Funding:** The author(s) received no specific funding for this work.

**Competing interests:** The authors have declared that no competing interests exist.

## Introduction

The North American Nursing Diagnosis Association International defines anxiety as "a vague and nuisance feeling of discomfort or fear". It is an apprehensive feeling caused by the anticipation of something that might happen, whether good or bad. Admission to a hospital, a surgical procedure, and anesthesia fear are events that can provoke anxiety [1,2]. It is a present feeling of around 80% of adult patients awaiting some surgery and its symptoms begin when the patient is informed that the surgical procedure is necessary, increasing during hospitalization and reaching its maximum peak before anesthesia [3]. One of the strategies to decrease the levels of anxiety during the preoperative period is welcoming and increasing patient autonomy through preoperative guidelines, which consists of providing information about health conditions and procedures to be performed, and the presence of a family member [4,5]. The autonomy principle is one of the medical bioethics pillars and the patient must have the power to make decisions about their treatment, increasing the power of choice in the preoperative period, helping to reduce anxiety related to the surgical procedure, and satisfaction with the coping process [4,5].

In most hospitals, patients are referred to the surgical center on a stretcher, accompanied by a nursing technician or nurse, wearing hospital's clothes (hospital pajamas). Studies showed that the patient's self-control and autonomy loss during the surgical center route, combined with poor eye contact and the patient's health condition, compromise satisfaction and increase anxiety in the preoperative period [6–8]. Studies carried out in different hospitals, with different populations, demonstrated that patients who walked to the surgical center felt calmer and more satisfied in the preoperative period. The stretcher use made them more anxious, distressed and even inferior since they remained lying down and drew more attention through the way [6,7]. When offered the choice about how to be referred to the surgical center, patients overall preferred to walk. Patients who chose to use a wheelchair or stretcher, when questioned about the choice, reported that they were concerned about the lack of dignity related to the hospital clothing, showing that it is possible to change the way how patients are leaded to the surgical center, allowing them to walk with family members and nursing staff [6,7,9,10].

In this context, walk to the surgical center for elective surgeries can reduce levels of anxiety, increase satisfaction, and bring independence feeling to the patient. Furthermore, this kind of conduction increases patient's autonomy, can improve hospital logistics and consequently result in a better postoperative recovery [6,7–10]. Therefore, could the type of conduction to the surgical center influence cancer patients' levels of anxiety and depression? To our knowledge, there are no studies evaluating the levels of anxiety and satisfaction of surgical cancer patients according to the way they are referred to the surgical center. The aim of this study was to evaluate the levels of anxiety, depression and satisfaction of cancer patients according to the type of conduction to the surgical center.

## Materials and methods

Randomized, open and parallel clinical trial, carried out at Barretos Cancer Hospital from January 2019 to August 2021. Were included patients diagnosed with cancer, between 18 and 70 years old, both genders, performance status (PS) classified as 0 and 1 [11], zero or low risk according to the falls risk assessment scale [12] and elective cancer surgery indication. Patients in the following situations were excluded: those who, moments before surgery, had an adverse event that could jeopardize walking to the surgical center, patients with physical disabilities or in need for walking assistance, postoperative Intensive Care Unit (ICU) indication, previously diagnosis of psychiatric disorders and anxiolytic and antidepressant medications use, and unaccompanied patients on the surgery day.

## Study conduction

Patients who were at the surgical inpatient unit reception waiting for the surgical procedure were screened for the study. Throughout medical notes and pre-anesthetic evaluation form it was possible to identify age, PS [11], falls risk factors, physical disability, daily medications, among other necessary information. After randomization, the nursing professional (stretcher bearer) was instructed about which group the patient was allocated. The patient was not informed about how she/he would be taken to the surgical center, for not inducing the questionnaires answers. The control group (stretcher) were received in their room by a nursing technician, their companion and the stretcher bearer, who conducted them to the surgical center, lying down on a stretcher, wearing hospital pajamas, with their companion. The experimental group (ambulation) left the surgical inpatient unit reception and were guided to the surgical center walking, with their own clothes, along with a companion and the stretcher bearer. At the surgical center arrival, all patients (both groups) answered the HAD scale questionnaire - anxiety and depression level assessment [13] and the Surgery Health Care Satisfaction Assessment Questionnaire (Sati-Cir). The second questionnaire was developed for the present study, to assess the patient's satisfaction with the hospital care, and to assess their satisfaction about the way they were conducted to the surgical center. Some steps were respected for the questionnaire elaboration: active search in literature; creation of 40 questions based on our institution's demands; analysis of the questions by a 10 researchers' specialist committee from our institution with questionnaires experience. After organizing the expert committee opinion and identifying the issues raised by them, 14 questions were selected. The development of the questionnaire was motivated by the lack of suitable instruments to assess patient satisfaction with healthcare services, as the existing questionnaires were inadequate to meet the specific objectives of the study. The collected data included baseline characteristics (age, weight, height, race, comorbidities, education) and clinical data (type of cancer, clinical staging, surgery date, type of surgery performed, Eastern Cooperative Oncology Group Performance Status (ECOG) [11] and history of previous surgeries).

The patient's study consent was carried out a posteriori, in other words, Informed Consent Form (ICF) application was carried out after the study procedures (conduction and questionnaires' application) to avoid bias related to the questionnaires' answers. The data collection process was as follows: participants proceeded to the surgical center, where they received a brief explanation about the anxiety, depression, and satisfaction questionnaires prior to the surgical procedure. They then completed the questionnaires, and after finishing them, the informed consent process was initiated, with a detailed explanation of the study and the reason for signing the consent form at a later time. The Informed Consent Form (ICF) was signed only if the participant agreed to take part in the study; otherwise, the questionnaire was immediately discarded in their presence. The objective was not to induce the patient to believe that being taken to the surgical center walking would be more comfortable and cause greater satisfaction. This project complies with the Guidelines and Standards Regulating Research Involving Human Beings and was approved by the Barretos Cancer Hospital Institutional Review Board (IRB) under the *CAAE: 82010218.2.0000.5437*, registered in Clinical trials: Barretos Cancer Hospital Protocol Record 1527/2018. The first patient was included after Clinical Trials approval.

## Statistical analysis

A sample size calculation was performed to compare proportions referring to the "Calm/mildly anxious" group, based on the results found by Kojima et al [7]. Considering a significance level ($\alpha$) of 0.05 and a power ($1 - \beta$) of 0.80, a sample size of 176 patients was

determined, with 88 patients allocated to the stretcher group and 88 patients allocated to the ambulation group. The statistical test used was Fisher's Exact Test (two-sided). Descriptive statistics were used to summarize demographic and clinical characteristics, through median, maximum and minimum values for continuous variables and relative frequencies for categorical variables. The Mann-Whitney Test was applied for quantitative variables, Chi-square or Fisher's exact tests were used for categorical variables to compare proportions between groups. The significance level adopted was 0.05. All data were stored and managed through REDCap database [14] and analysis was conducted using SPSS software (Statistical Package for Social Sciences) version 27.

The randomization was generated through the platform REDCap [14], creating a random list for study entry. The theoretical model used was a comparison between groups, and the randomization method was simple randomization without matching.

## Results

Initially, 239 patients were screened, of whom 45 were deemed ineligible for participation. Randomization was performed with 194 patients, of whom 16 were unable to complete the questionnaires and sign the consent form, as they were taken to the surgical center before the research team could carry out these procedures. Additionally, there were 2 refusals to participate. One hundred and seventy-six patients were included. Thus, 94 patients were taken to the surgical center on a stretcher, and 82 patients walked. The CONSORT (Consolidated Standards of Reporting Trials) flow diagram [15] is shown in Fig 1. Table 1 shows the sociodemographic clinical/surgical and tumor characteristics of the included patients.

The patients' median age was 44.9 years (26.3–68.4) in the ambulation group and 50.6 years (20.4–69.9) in the stretcher group (p = 0.048). Among the ambulation group patients, 62 (75.6%) were women and 20 (24.4%) were men, and in the stretcher group, 61 (64.9%) were women and 33 (35.1%) were men (p = 0.122). In terms of education, completed secondary or technical education or incomplete university was present in 34 (44.2%) patients in the ambulation group and in 30 (34.1%) patients in the stretcher group, (p = 0.142). Performance Status 0 and 1 was present in 70 (85.4%) and 12 (14.6%) patients in the ambulation group while in the stretcher group was 81 (86.2%) and 13 (13.8%), (p = 0,879).

Breast cancer was the most frequent type of cancer in both groups, 28 (34.1%) patients in the ambulation group and 32 (34.0%) in the stretcher group (p = 0.461). Laparotomy surgery was performed in 67 patients (82.7%) in the ambulation group and 72 patients (78.3%) in the stretcher group (p = 0.740). Median number of surgical procedures previously performed by each patient in both groups was 2, ranging from 1 to 7 (p = 0.391).

There was no statistical difference in anxiety and depression levels between groups. Fifteen (18.3%) patients in the ambulation group and 20 (21.3%) patients in the stretcher group were anxious (p = 0.621). Regarding the assessment of depression levels, 6 (7.3%) patients in the walking group and 4 (4.3%) patients in the stretcher group met depression criteria (p = 0.518) (Table 2).

Data from Surgery Health Care Satisfaction Assessment Questionnaire (Sati-Cir) are shown in Table 3. Regarding the way the patient was taken to the surgical center, there was a statistical difference related to the surgery waiting time, in the ambulation group 75 (91.5%) patients were very satisfied or satisfied, 3 (3.7%) patients were indifferent, and 4 (4.9%) patients were dissatisfied or very dissatisfied, however, in the stretcher group, 70 (74.5%) patients were very satisfied or satisfied, 16 (17.0%) were indifferent, and 8 (8.5%) were dissatisfied or very dissatisfied (p = 0.008). There was also a difference related to the satisfaction with the clothing used to be taken to the surgical center, 51 (100%) patients who walked around wearing their own clothes were very satisfied or satisfied, while in the stretcher

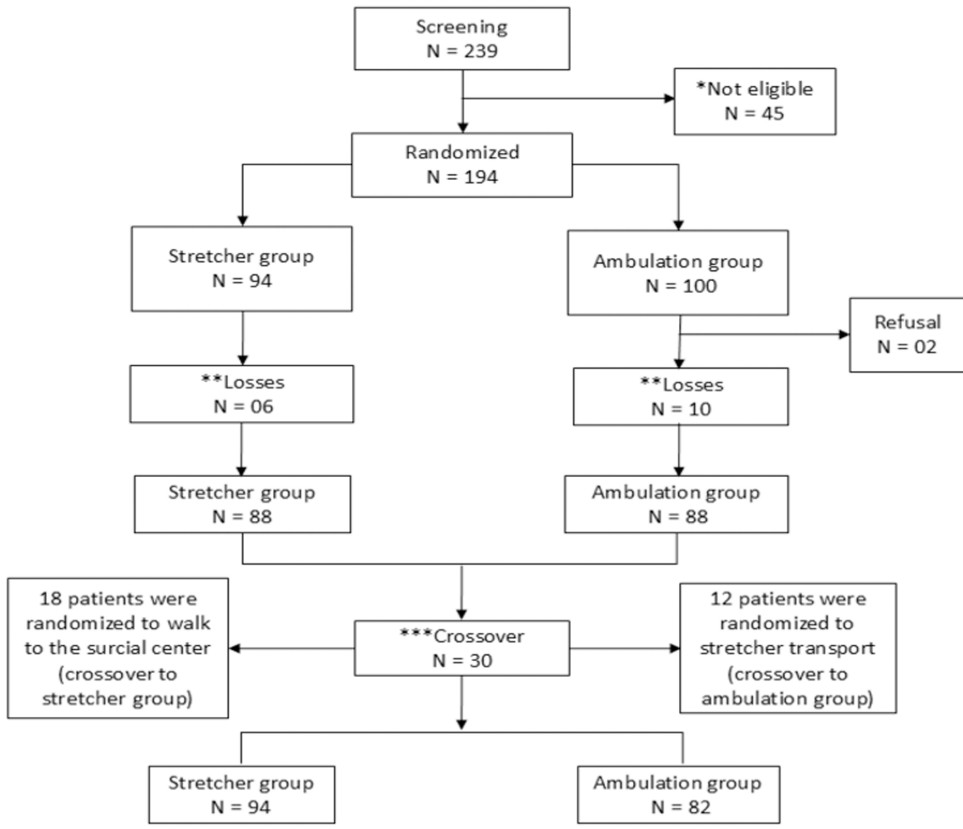

**Fig 1. The CONSORT flow diagram. * Not eligible reasons** - 5 patients were over 70 years of age. - 8 patients required assistance to ambulate. - 4 patients had cognitive impairment. - 10 patients were using antidepressant/anxiolytic medications. - 18 patients were at moderate/high risk for falls. **** Reasons for Losses**. All patients, from both groups, were referred to the surgical center to undergo the procedure before completing the questionnaires to assess anxiety and depression, and before signing the informed consent form. ***** Crossover** Logistical challenges led to 30 participants deviating from randomization: 12 assigned to stretcher group walked instead, and 18 assigned to ambulation group were transported by stretcher. As the study focused on satisfaction, anxiety, and depression based on actual transport, analysis was conducted accordingly.

group, 49 (79.0%) patients were very satisfied or satisfied with the clothing (p = 0.001). When questioned about the option of choosing how to be taken to the surgical center after going there, in the ambulation group, 80 (97.6%) patients would opt for walking to the surgical center, 1 (1.2%) patient would choose to be transported on a stretcher, and 1 (1.2%) patient would choose to be transported in a wheelchair. In the stretcher group, 44 (46.8%) patients would opt for walking to the surgical center, 49 (52.1%) would choose to be transported on a stretcher and 1 (1.1%) patient would choose to be transported in a wheelchair (p < 0.001).

## Discussion

The main objective was to evaluate the impact of walking to the surgical center in cancer patients, accompanied by a family member, wearing their own clothes. Regarding the primary outcome, no statistical difference was identified related to anxiety and depression symptoms according to the way they were taken to the surgical center, in both groups, most patients did not have anxiety and depression symptoms. However, when patient's satisfaction was evaluated, 100% of those who were taken to the surgical center walking, wearing their own clothes

**Table 1. Sociodemographic, clinical/surgical and tumor characteristics.**

| Variable | Stretcher (94) | Ambulation (82) | Total | P-value |
|---|---|---|---|---|
| | N* (%) | N* (%) | | |
| Age (Years) Median (Min-Max) | 50.6 (20.4-69.9) | 44.9 (26.3-68.4) | 176 (100%) | 0.048 |
| Ethnicity | | | | |
| White | 70 (76.1%) | 62 (78.5%) | 132 (77.2%) | 0.710 |
| Not White | 22 (23.9%) | 17 (21.5%) | 39 (22.8%) | |
| Sex | | | | |
| Male | 33 (35.1%) | 20 (24.4%) | 53 (30.1%) | 0.122 |
| Female | 61 (64.9%) | 62 (75.6%) | 123 (69.9%) | |
| Education | | | | |
| Illiterate | 1 (1.1%) | 2 (2.6%) | 3 (1.8%) | 0.142 |
| CRW/ IPE | 24 (27.3%) | 16 (20.8%) | 40 (24.2%) | |
| CPE/ ISE | 18 (20.5%) | 7 (9.1%) | 25 (15.2%) | |
| CSE/ IU | 30 (34.1%) | 34 (44.2%) | 64 (38.8%) | |
| CU/ PG | 15 (17.0%) | 18 (23.4%) | 33 (20.0%) | |
| **PS ECOG** | | | | |
| PS 0 | 81 (86.2%) | 70 (85.4%) | 151 (85.8%) | 0.879 |
| PS 1 | 13 (13.8%) | 12 (14.6%) | 25 (14.2%) | |
| Risk of falls | | | | |
| Risk-free | 61 (64.9%) | 59 (72.0%) | 120 (68.2%) | 0.316 |
| Low risk | 33 (35.1%) | 23 (28.0%) | 56 (31.8%) | |
| Type of Cancer | | | | |
| Breast | 32 (34.0%) | 28 (34.1%) | 60 (34.1%) | 0.461 |
| Ovary | 3 (3.2%) | 4 (4.9%) | 7 (4.0%) | |
| Cervix | 5 (5.3%) | 1 (1.2%) | 6 (3.4%) | |
| Endometrium | 0 (0.0%) | 1 (1.2%) | 1 (0.6%) | |
| Head and neck | 6 (6.4%) | 7 (8.5%) | 13 (7.4%) | |
| Prostate | 6 (6.4%) | 4 (4.9%) | 10 (5.7%) | |
| Genitourinary | 11 (11.7%) | 3 (3.7%) | 14 (8.0%) | |
| Gastrointestinal | 2 (2.1%) | 3 (3.7%) | 5 (2.8%) | |
| Non-melanoma skin | 5 (5.3%) | 9 (11.0%) | 14 (8.0%) | |
| Melanoma | 3 (3.4%) | 3 (3.7%) | 6 (3.4%) | |
| Others | 21 (22.3%) | 19 (23.2%) | 40 (22.7%) | |
| Surgical access | | | | |
| Laparotomy | 72 (78.3%) | 67 (82.7%) | 139 (80.3%) | 0.740 |
| Laparoscopic | 18 (19.6%) | 12 (14.8%) | 30 (17.3%) | |
| Robotics | 2 (2.2%) | 2 (2.5%) | 4 (2.3%) | |
| Surgical procedures per patient median number | | | | |
| Previous surgical procedures per patient | 2 (1-5) | 2 (1-7) | | 0.391 |

This is the Table 1 legend.

(*) Cases with missing data were excluded from the analysis.

CRW = Can Read and Write/ IPE = Incomplete Primary Education

CPE = Completed Primary Education/ ISE = Incomplete Secondary Education

CSE = Completed Secondary Education/ IU = Incomplete University

CU = Complete University/ PG = Postgraduation

ECOG = Performance Status Scale

PS 0 = Normal activity

PS 1 = Disease symptoms, but walk and carry out normal daily activities

**Table 2. HAD Scale – Level of Anxiety and Depression Assessment according to the type of transport to the surgical center.**

| ANXIETY AND DEPRESSION SCORE | STRETCHER (94) | AMBULATION (82) | TOTAL | P-VALUE |
|---|---|---|---|---|
| | N* (%) | N* (%) | | |
| NO ANXIETY (0–8) | 74 (78.7%) | 67 (81.7%) | 141 (80.1%) | 0.621 |
| ANXIETY ≥ 9 | 20 (21.3%) | 15 (18.3%) | 35 (19.9%) | |
| NO DEPRESSION (0–8) | 90 (95.7%) | 76 (92.7%) | 166 (94.3%) | 0.518 |
| DEPRESSION ≥ 9 | 4 (4.3%) | 6 (7.3%) | 10 (5.7%) | |

This is the Table 2 legend.

(*) Cases with missing data were excluded from the analysis.

**Table 3. Surgery Health Care Satisfaction Assessment Questionnaire (Sati-Cir).**

| Variable | Stretcher (94) | Ambulation (82) | Total | P-value |
|---|---|---|---|---|
| | N* (%) | N* (%) | | |
| **What is your level of satisfaction related to the clarification of your disease's doubts?** | | | | |
| VS/ S | 85 (90.4%) | 78 (95.1%) | 163 (92.6%) | 0.306 |
| Indifferent | 2 (2.1%) | 2 (2.4%) | 4 (2.3%) | |
| D/ VD | 7 (7.4%) | 2 (2.4%) | 9 (5.1%) | |
| **What is your level of satisfaction related to your participation in decisions related to your treatment?** | | | | |
| VS/ S | 92 (97.9%) | 81 (98.8%) | 173 (98.3%) | >0.999 |
| Indifferent | 2 (2.1%) | 1 (1.2%) | 3 (1.7%) | |
| D/ VD | 0 (0.0%) | 0 (0.0%) | 0 (0.0%) | |
| **What is your level of satisfaction related to the opportunities given to you by the team to ask questions?** | | | | |
| VS/ S | 92 (97.9%) | 81 (98.8%) | 173 (98.3%) | 0.349 |
| Indifferent | 2 (2.1%) | 0 (0.0%) | 2 (1.1%) | |
| D/ VD | 0 (0.0%) | 1 (1.2%) | 1 (0.6%) | |
| **What is your level of satisfaction related to how you get information when you ask for it?** | | | | |
| VS/ S | 91 (96.8%) | 80 (97.6%) | 171 (97.2%) | 0.511 |
| Indifferent | 2 (2.1%) | 0 (0.0%) | 2 (1.1%) | |
| D/ VD | 1 (1.1%) | 2 (2.4%) | 3 (1.7%) | |
| **What is your level of satisfaction related to the nursing team care to preserve your privacy?** | | | | |
| VS/ S | 93 (98.9%) | 82 (100.0%) | 175 (99.4%) | >0.999 |
| Indifferent | 1 (1.1%) | 0 (0.0%) | 1 (0.6%) | |
| D/ VD | 0 (0.0%) | 0 (0.0%) | 0 (0.0%) | |
| **What is your level of satisfaction related to the waiting time to be hospitalized?** | | | | |
| VS/ S | 76 (80.9%) | 65 (79.3%) | 141 (80.1%) | 0.481 |
| Indifferent | 7 (7.4%) | 10 (12.2%) | 17 (9.7%) | |
| D/ VD | 11 (11.7%) | 7 (8.5%) | 18 (10.2%) | |
| **What is your level of satisfaction related to the waiting time to the surgery?** | | | | |
| VS/ S | 70 (74.5%) | 75 (91.5%) | 145 (82.4%) | *0.008* |
| Indifferent | 16 (17.0%) | 3 (3.7%) | 19 (10.8%) | |
| D/ VD | 8 (8.5%) | 4 (4.9%) | 12 (6.8%) | |

*(Continued)*

**Table 3.** (Continued)

| Variable | Stretcher (94) | Ambulation (82) | Total | P-value |
|---|---|---|---|---|
| | N* (%) | N* (%) | | |
| *What is your level of satisfaction related to how you were taken to the surgical center?* | | | | |
| VS/ S | 89 (94.7%) | 81 (98.8%) | 170 (96.6%) | 0.374 |
| Indifferent | 4 (4.3%) | 1 (1.2%) | 5 (2.8%) | |
| D/ VD | 1 (1.1%) | 0 (0.0%) | 1 (0.6%) | |
| *What is your level of satisfaction related to the clothes you were wearing when were taken to the surgical center?* | | | | |
| VS/ S | 49 (79.0%) | 51 (100.0%) | 100 (88.5%) | *0.001* |
| Indifferent | 7 (11.3%) | 0 (0.0%) | 7 (6.2%) | |
| D/ VD | 6 (9.7%) | 0 (0.0%) | 6 (5.3%) | |
| *Do you believe that your dignity was preserved related to the clothes you were wearing when were taken to the surgical center?* | | | | |
| Yes | 59 (95.2%) | 51 (100%) | 110 (97.3%) | 0.250 |
| No | 3 (4.8%) | 0 (0,0%) | 3 (2.7%) | |
| *Do you think that your autonomy was preserved related to the way you were taken to the surgical center?* | | | | |
| Yes | 90 (95.7%) | 82 (100%) | 172 (97.7%) | 0.124 |
| No | 4 (4.3%) | 0 (0,0%) | 4 (2.3%) | |
| *If you could choose how to be taken to the surgical center, which would it be?* | | | | |
| Stretcher | 49 (52.1%) | 1 (1.2%) | 50 (28.4%) | *< 0.001* |
| Wheelchair | 1 (1.1%) | 1 (1.2%) | 2 (1.1%) | |
| Ambulation | 44 (46.8%) | 80 (97.6%) | 124 (70.5%) | |

This is the Table 3 legend.

(*) Cases with missing data were excluded from the analysis.

VS = Very Satisfied/ S = Satisfied

D = dissatisfied/ VD = very dissatisfied

and in the company of a family member, were very satisfied or satisfied. Although, in the group conducted on a stretcher and wearing pajamas, this number drops to 79.0% (p-value 0.001). Among ambulation group patients, 91.5% were also satisfied or very satisfied when asked about the waiting time for the surgery (p-value 0.008). Nonetheless, this evaluation may be a consequence of the fact that 97.6% of the patients who walked were very satisfied about the way they were taken.

Other studies had previously analyzed the impact of the type of conduction to the surgical center. Kojima et al. [7] performed one of the first reports about the possibility of this change in healthcare practice. Their study hypothesized that entering the surgical center lying on a stretcher would increase anxiety, so they investigated the effects of walking to the operating room compared to be transported on a stretcher on anxiolytic premedication. The results demonstrated that patients were less anxious in the ambulation group, whereas in our population of cancer patients, there was no difference between groups (stretcher vs ambulation) related to anxiety symptoms and no patient in our trial had anxiolytic medication before surgery. Majumdar et al.[16] carried out a retrospective cohort study with patients undergoing outpatient cancer surgery, and highlighted that 20% of the patients had significant levels of preoperative anxiety. The population majority consisted of women undergoing breast reconstruction surgery, and the results showed that breast cancer patients had preoperative anxiety symptoms. In our clinical trial, most of patients were women who underwent surgery for breast cancer treatment (34.1%), however when we evaluated the anxiety symptoms, we identified that, unlike Majumdar study findings, 91.7% of our patients with breast tumors were not

anxious. It is important to emphasize that in our study we evaluated the anxiety symptoms using the HADS scale [13], consolidated worldwide. In the study mentioned above, the assessment of anxiety was performed retrospectively by clinical assessment through changes in vital signs or verbalization of anxiety.

At the present study, the patients who were taken to the surgical center walking around, waited for the surgery with their relatives at the reception of the surgical wards. We believe that waiting in the company of others in similar situation, and the experiences exchange, culminated in greater satisfaction regarding the waiting time for the surgical procedure, where 91.5% of the patients were very satisfied or satisfied (p-value 0.008). Patients in the stretcher group were taken to their hospital bed and waited for the surgery in the room with their family member. In this group, satisfaction with the waiting time for surgery was 74.5%. Satisfaction regarding clothing was an important point in the study findings, 100% of the patients who were taken to the surgical center walking and wearing their own clothes were very satisfied or satisfied (p-value 0.001). However, when asked about dignity related to clothing, both groups (stretcher vs/and ambulation) believed that it had been preserved. Unlike the results found by Nagraj et al.[6] who identified the patients' concern with the lack of dignity associated with the hospital nightgown (where the opening is on the back), we believe that the preservation of dignity, referred by both groups evaluated in our study, is related to the fact that the patient included in the stretcher group was also dressed in a hospital nightgown, but he was lying on a stretcher and covered with a sheet throughout the entire path. We believe that if the patient were offered the opportunity to walk to the operating room wearing this suit, the results would probably be different. Maharjan et al.[8] carried out a cross-sectional study aiming to demonstrate the feasibility of walking to the surgical center. Like our study, patients were referred to the surgical center with their family member and nursing professional. As a different aspect, to the patients were offered the opportunity to walk to the surgical center, while in our study, the way of conduction was designated through randomization. In the Maharjan study, there is no information on whether there was any screening for the risk of falls. All patients were invited to walk and 97% of them were classified as non-anxious, but anxiolytic medication was prescribed the day before surgery, which may have influenced the high rate of those classified as non-anxious. In Kojima et al.[7] study anxiolytic medication was also used prior to the surgical procedure, and they referred to it as a difficulty in assessing anxiety, due to the fact that the patients who were transported to the surgical center on a stretcher were sleeping. Our study followed the recommendation of the European group ERAS (Enhanced Recovery After Surgery), which suggests against the use of anxiolytic medications before surgical procedures, which could affect the World Health Organization checklist routines of safe surgery procedures [17].

The reviewed studies were observational, and the patients chose how to be conducted to the surgical center according to their preference. We found that only orthopedic patients with surgery on the lower limbs, elderly patients with poor mobility and those concerned with the dignity related to clothing, choose to be transported to the surgical center on a stretcher, the others, as well as in our study, preferred to be conducted to the surgical center walking (5, 8–10). The option of changing this care practice, in addition to offering greater autonomy and freedom of choice for patients included on the category with no risk of walking, also optimizes the work of the nursing team, since the professional stretcher bearer is able to conduct more than one patient at a time without offering risks, as they are in the company of their family member for any support during the journey.

As strengths of the study, we emphasize the study design, methodological rigor, and strategies to reduce possible biases. To our knowledge, we believe that this is the first randomized clinical trial evaluating the impact on levels of anxiety and satisfaction related to the

way oncologic surgical patients were taken to the surgical center. It was possible to guarantee methodological rigor in the study development, because it was conducted by a specialized and duly trained team. The institution counts with a research initiative (investigator-initiated) study specialized department for over 10 years. A differential of this study, compared to those previously published, is that the patients who were conducted to the surgical center walking, made the route with a family member and wearing their own clothes, changing to hospital nightgown only after surgical center admission. Another strength is the fact that in our study population, no patient received anxiolytic medication on the day before surgery. Another strategy used aiming to reduce the study bias was the application of the consent informed form after completing the route and the study questionnaires, in order to not influence the patient to believe that walking would be the best option.

It is important to point out some limitations of the study, such as the patients' levels of anxiety at the interview, which may be related to the non-inclusion of patients with major surgery and ICU postoperative recovery need. Only PS 0 and 1 patients were included. Furthermore, our institution only offers public health treatment, prioritizing humanization and focusing on the patient's need, that can be a possible confounding factor. That fact could cause the feeling of gratitude on patients, which could have an impact on all patient satisfaction assessments. Among those patients included in the study, 41.2% had low educational level and 40% were classified as incomplete primary and secondary education. We believe that the low educational level may have hindered the understanding of the questionnaire and the self-perception of anxiety by the patients, since low education level is directly related to the neediest population. Another fact that deserves emphasis is the non-inclusion of patients previously diagnosed with anxiety disorder. We emphasize that the non-inclusion of those patients was intended to identify whether or not the act of walking to the surgical center influenced the symptoms of anxiety.

As future perspectives, we highlight the importance of conducting studies evaluating anxiety symptoms according to the type of transportation to the surgical room in patients undergoing all types of surgical procedures, as well as levels of pre-diagnosed anxiety. Furthermore, it is essential to include a larger sample size and more diverse populations, as the present study was conducted exclusively with Brazilian participant's users of the public health system. We also hope to change the routine of patient's transport to the surgical center, since the study showed that it is a beneficial practice, which does not offer risks to the patient, optimizes the routine of the nursing team and provides greater well-being to the patient in moments that precede the surgery.

In our hospital, as in most hospitals worldwide, patients are transported on stretchers and in hospital-provided pajamas due to an outdated routine that is practical and seldom questioned. There is no reason for patients with performance status 0 and 1, and no fall risk, to be transported on stretchers and in hospital pajamas, which convey a sense of dependency and illness. Based on the study findings and other evidence from the literature, institutions should reconsider the mode of transport for patients with performance status 0 and 1, and no fall risk, when transferring them to the surgical center.

## Conclusions

It was possible to conclude that there was no reduction in preoperative anxiety and depression symptoms according to the way the patients were driven to the surgical center. However, changing this practice directly influenced patients' satisfaction, as those who were taken to the surgical center walking were more satisfied with the surgery waiting time and more satisfied with their clothing. In addition, when all the patients were questioned about how they would

like to be taken to the surgical center, 70.5% of them reported that they would choose to walk, wearing their own clothes and in the company of their family member, reaffirming the importance of implementing this practice in patient care.

## Supporting information

**S1 Protocol. Protocol English.**
(DOCX)

**S2 Protocol. Protocol Portuguese.**
(DOCX)

**S3 CONSORT checklist.**
(DOC)

## Acknowledgments

We are thankful to Welinton Yoshio Hirai for the statistical revision and new statistical analysis.

This work was supported by the Barretos Cancer Hospital.

## Author contributions

**Conceptualization:** Gabriela da Silva Oliveira, Ricardo dos Reis.

**Data curation:** Gabriela da Silva Oliveira, Vinicius Bars da Silva, Ricardo dos Reis.

**Formal analysis:** Gabriela da Silva Oliveira, Marcos Alves de Lima, Ricardo dos Reis.

**Methodology:** Gabriela da Silva Oliveira, Marcos Alves de Lima, Ricardo dos Reis.

**Project administration:** Gabriela da Silva Oliveira, Vinicius Bars da Silva, Ricardo dos Reis.

**Supervision:** Ricardo dos Reis.

**Writing – original draft:** Gabriela da Silva Oliveira, Paula Moreira da Silva Sabaini, Priscila Grecca Pedrão, Vinicius Bars da Silva, Marcos Alves de Lima, Thiago Emilio da Silva Mantovani, Carlos Eduardo Mattos da Cunha Andrade, Ricardo dos Reis.

**Writing – review & editing:** Gabriela da Silva Oliveira, Paula Moreira da Silva Sabaini, Priscila Grecca Pedrão, Vinicius Bars da Silva, Marcos Alves de Lima, Thiago Emilio da Silva Mantovani, Carlos Eduardo Mattos da Cunha Andrade, Ricardo dos Reis.

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
