## [Decision Letter · Decision Letter 0]

22 Oct 2024

PONE-D-24-33555Emotional impact according to the way conducting cancer patients to the surgical center: a randomized clinical trial comparing ambulation to the stretcherPLOS ONE

Dear Dr. Oliveira,

Thank you for submitting your manuscript to PLOS ONE. After careful consideration, we feel that it has merit but does not fully meet PLOS ONE’s publication criteria as it currently stands. Therefore, we invite you to submit a revised version of the manuscript that addresses the points raised during the review process.

We look forward to receiving your revised manuscript.

Kind regards,

Alexandre Bonatto

Academic Editor

PLOS ONE

Journal Requirements:

1. When submitting your revision, we need you to address these additional requirements. Please ensure that your manuscript meets PLOS ONE's style requirements, including those for file naming. The PLOS ONE style templates can be found at https://journals.plos.org/plosone/s/file?id=wjVg/PLOSOne_formatting_sample_main_body.pdf and https://journals.plos.org/plosone/s/file?id=ba62/PLOSOne_formatting_sample_title_authors_affiliations.pdf 2. We note that the original protocol file you uploaded contains a confidentiality notice indicating that the protocol may not be shared publicly or be published. Please note, however, that the PLOS Editorial Policy requires that the original protocol be published alongside your manuscript in the event of acceptance. Please note that should your paper be accepted, all content including the protocol will be published under the Creative Commons Attribution (CC BY) 4.0 license, which means that it will be freely available online, and any third party is permitted to access, download, copy, distribute, and use these materials in any way, even commercially, with proper attribution. Therefore, we ask that you please seek permission from the study sponsor or body imposing the restriction on sharing this document to publish this protocol under CC BY 4.0 if your work is accepted. We kindly ask that you upload a formal statement signed by an institutional representative clarifying whether you will be able to comply with this policy. Additionally, please upload a clean copy of the protocol with the confidentiality notice (and any copyrighted institutional logos or signatures) removed. 3. We note that your Data Availability Statement is currently as follows: [All relevant data are within the manuscript and its Supporting Information files.] Please confirm at this time whether or not your submission contains all raw data required to replicate the results of your study. Authors must share the “minimal data set” for their submission. PLOS defines the minimal data set to consist of the data required to replicate all study findings reported in the article, as well as related metadata and methods (https://journals.plos.org/plosone/s/data-availability#loc-minimal-data-set-definition). For example, authors should submit the following data: - The values behind the means, standard deviations and other measures reported;- The values used to build graphs;- The points extracted from images for analysis. Authors do not need to submit their entire data set if only a portion of the data was used in the reported study. If your submission does not contain these data, please either upload them as Supporting Information files or deposit them to a stable, public repository and provide us with the relevant URLs, DOIs, or accession numbers. For a list of recommended repositories, please see https://journals.plos.org/plosone/s/recommended-repositories. If there are ethical or legal restrictions on sharing a de-identified data set, please explain them in detail (e.g., data contain potentially sensitive information, data are owned by a third-party organization, etc.) and who has imposed them (e.g., an ethics committee). Please also provide contact information for a data access committee, ethics committee, or other institutional body to which data requests may be sent. If data are owned by a third party, please indicate how others may request data access. 4. Please include captions for your Supporting Information files at the end of your manuscript, and update any in-text citations to match accordingly. Please see our Supporting Information guidelines for more information: http://journals.plos.org/plosone/s/supporting-information.

Additional Editor Comments:

I apologize for the time to conclude the review. It has been quite challenging to secure the reviewers (additionally, some of them asked for an extended period to conclude their reviews). 

Reviewers' comments:

Reviewer's Responses to Questions

**Comments to the Author**

1. Is the manuscript technically sound, and do the data support the conclusions?

Reviewer #1: Yes

Reviewer #2: Partly

Reviewer #3: Partly

Reviewer #4: Partly

2. Has the statistical analysis been performed appropriately and rigorously?

Reviewer #1: Yes

Reviewer #2: No

Reviewer #3: Yes

Reviewer #4: I Don't Know

3. Have the authors made all data underlying the findings in their manuscript fully available?

Reviewer #1: Yes

Reviewer #2: Yes

Reviewer #3: Yes

Reviewer #4: Yes

4. Is the manuscript presented in an intelligible fashion and written in standard English?

Reviewer #1: Yes

Reviewer #2: Yes

Reviewer #3: No

Reviewer #4: No

5. Review Comments to the Author

Reviewer #1: Under review of the article entitled: “Emotional impact according to the way conducting cancer patients to the surgical center: a randomized clinical trial comparing ambulation to the stretcher.”, whose main objective was to evaluate the impact of walking to the surgical center in cancer patients, accompanied by a family member, wearing their clothes. An interesting article was observed, but it needs an in-depth review. This is a critical study that is studying the emotional issues of individuals with cancer, in addition to proposing a humanized practice for their care. Some additional clarifications are essential so we can be sure of the quality of the study: I did not understand why there were no losses in the study. A flowchart is important. What was the strategy to have 100% of people participating in the study? Furthermore, there were no complications, and no one died or did not want to participate in the study. Although the authors explain the reason for not applying the questionnaires before the informed consent form so as not to bias the responses to the questionnaires analyzed, the study was approved by the Hospital's Research Ethics Committee, and its protocol was registered with Clinical Trials, I would like more information to be added. Was the person exposed to a questionnaire without knowing they would participate in a research study? This did not influence the selection of participants. It would not be necessary to have more information about the person's health conditions and to control for possible confounding factors such as participants in psychotherapy with a diagnosis of generalized anxiety disorder and others. What would be the theoretical model of the study? The study was randomized in RedCap. What method was used? Simple random? Was there any criterion or pairing? Explain in greater detail the blinding of the study. Many of the Consort items were presented as NA. I want the authors to be able to provide more details on why the study does not fit the item requested as necessary for the study to be of good quality.

Reviewer #2: This manuscript presents data analysis from a randomized control trial (RCT) to compare anxiety, depression and satisfaction among cancer patients, according to the type of route to the surgical center. The topic is of importance, and the study was registered as a RCT within the clinicaltrials.gov (with a valid NCT number), and was approved by the respective IRB/Ethics Committee. While the study objectives sound interesting, is important, and on target, some shortcomings were observed in regards to abiding by the CONSORT guidelines for conducting and reporting results of high-quality randomized controlled trials (RCTs). Some other (statistical) comments were also provided.

1. Methods:

Methods reporting need some work. An orderly manner is suggested, following CONSORT guidelines, without repeating information, such as Trial Design, Participant Eligibility Criteria and settings, Interventions, Outcomes, sample size/power considerations, Interim analysis and stopping rules, Randomization (details on random number generation, allocation concealment, implementation), Blinding issues, etc, should be mentioned. The authors are advised to create separate subsections for each of the possible topics (whichever necessary), and that way produce a very clear writeup. They are advised to write it carefully, following nice examples in the manuscript below:

https://www.sciencedirect.com/science/article/pii/S0889540619300010

Specific comments:

(a) For instance, the randomization and allocation concealment should be made very clear (they are NOT the same thing); the trial staff recruiting patients should NOT have the randomization list. Randomization should be prepared by the trial statistician, and he/she would not participate in the recruiting.

(b) More details on the randomization is needed; just saying randomization was conducted via RedCap appeared half-hearted. For example, was a block randomization conducted?

(c) Sample size: The sample size/power statement should mention the "name of the specific test used", whether it's a 2-sided, or 1-sided, and the corresponding effect size desired. Again, a separate subsection requested.

(d) Statistical Analysis:

(d1) While assessing differences between the groups (stretcher vs ambulation), Mann-Whitney test was used. Please cite reasons, why this was used over usual t-tests. Likely, due to suspected Non-Gaussian behavior?

(d2) It's not clear, why there was never an attempt to assess the differences between the groups, "controlled for other important covariates", via a suitable regression approach. This would have made the analysis much more robust.

2. Results & Conclusions:

(a) The authors should check that any statement of significance should be followed by a p-value in the entire Results section. Otherwise, the Results section look OK; it's pretty straightforward.

(b) Conclusions should state that the current findings are ONLY based on the random samples derived from a Brazilian population, and should allude to future studies with much larger sample sizes and collected at other geographical areas to confirm the effectiveness of ambulation to stretcher.

Reviewer #3: This manuscript explores an interesting premise—the impact of different methods of patient transport to the surgical center on anxiety, depression, and satisfaction in cancer patients.

Both the problem addressed and the results seem to have limited relevance to current medical practice, especially in the field of quality of life research in cancer patients. The mode of patient transport is usually dictated by the patient’s general health status and not by personal preference. The practical significance of this study’s findings may therefore be limited, as the decision on how a patient is conducted to surgery often involves medical necessity rather than patient choice.

Although the study begins with an interesting premise, the results fail to deliver any novel or clinically meaningful insights. The findings that there is no significant difference in anxiety and depression levels between patient groups is unsurprising, given that the mode of transportation is unlikely to have a strong psychological impact in the context of surgery.

Reviewer #4: This is an interesting manuscript about an RCT investigating the emotional (depression and anxiety) impact of mode to transport (stretcher with patient gown or walking with own clothes) to surgery. Satisfaction was also assessed. The introduction is good, and the discussion section compares the study results with what is available in the literature. Some clarifications regarding the methods section would be helpful, and I also have some questions about the study rationale and choice of design (major issues described below). The manuscript would also benefit from proofreading as there are some issues regarding translations, grammar and structure. I have outlined some of these (but not all) as minor issues below.

Major issues

1. Materials and methods, lines 107-108: please provide more information on what type of randomisation was carried out in REDCap. Also consider rewriting “randomization occurred randomly” to reduce redundancy.

2. Materials and methods, lines 124-125. Consider rewriting this to highlight that available questionnaires were not sufficient to meet the study aims, as satisfaction questionnaires on their own are not really scarce in the literature (see a review here for an example): https://pmc.ncbi.nlm.nih.gov/articles/PMC10001171/pdf/healthcare-11-00639.pdf

3. Materials and methods (overall): Please add information on which outcomes were primary/secondary, clarify blinding (even if it is just stating that blinding was not possible due to the nature of the study).

4. Materials and methods, lines 129-137: The study has received ethical approval, but informed consent was obtained after participating in the study. I understand why this was needed, but I imagine some information was provided to patients (i.e. they were not just given questionnaires to complete without receiving any preliminary information). If this is the case, what information was provided? If this was not the case, were participants allowed to withdraw their data? I could not find this information in the original protocol (Portuguese).

5. Materials and methods, lines 138-140: More information on sample size calculation would be helpful. What proportions are being considered, for what measures?

6. Results, Table 1: Considering the ethnic mix in Brazil, where under 50% of the population is White (according to the Census), it is important to mention something in the discussion about the generalisability of study results. Why do the authors think most participants were White? Are there any implications?

7. Results, Table 1: While the conclusion (Abstract) mentions “no risk of falling”, Table 1 shows that for almost a third of all patients (over a third in the stretcher group) there was low risk of falling (as opposed to risk-free or no-risk). Consider rewriting the text in the abstract for consistency.

8. Methods and Results section (including Figure 1): more information is needed on recruitment (was it consecutive), and why there was a different number of participants in each group. What did not happen as expected in the protocol (i.e. 88 per group)? Figure 1 shows that 30 did not follow randomisation. Further information on this is also needed. How many from which group were not randomised, and why? Was this due to refusal in being conducted by stretcher/walking? How did 88 participants in the stretcher group became 94 after 30 of them did not follow randomisation? And importantly, what are the implications for the study results?

9. Results, page 8: Satisfaction with waiting time – could the authors clarity why this would be related to mode of transport? Does it take longer to go one way or the other? If so, how much longer, for which mode of transport? Adding this type of information would strengthen the manuscript.

10. It is very positive that participants were asked about their preferences, although this had to be done after they were conducted to surgery (this experience and the mode of transport may have influenced how they answered about their preferences). Consider discussing this.

11. Table 3. Was it clear in the questionnaire that the waiting time refers to waiting time in the hospital, as opposed to waiting time between knowing surgery was needed/booking the surgery and getting an appointment?

12. Table 3, missing data – I can see that there is a lot of missing data for satisfaction related to clothes, dignity being preserved and preferences, while there is no missing data for any of the other questions. Why do the authors think this is the case?

13. Discussion, lines 205-206: Consider clarifying this sentence, even if mentioning the outcomes subsequently. Furthermore, in this paragraph it is stated that anxiety and depression were primary outcomes. This does not fully match what is in the protocol in Portuguese (where only anxiety is described as the primary outcome). As before, please clarify in the methods section what were the primary and what were the secondary outcomes. If there were any deviations from the protocol, please state these in the manuscript, and briefly mention why there were deviations, and whether there are any implications for the study results.

14. Overall: Are there any implications of not having fully equivalent groups? In other words, could preferences and satisfaction have been different if those on a stretcher were allowed to wear their own clothes? Is the choice of mode of transport/clothing done based on usual hospital procedures? What is considered standard care in such circumstances? Consider approaching this issue in the discussion section.

15. Overall: I am struggling a bit to understand if it is appropriate to use a scale for depression after such a short intervention. Is it possible to attribute depression to the mode of transport (also bearing in mind we do not know how patients would have scored at baseline – this also applies to anxiety, although I am more comfortable with anxiety being measured)? I am happy to be convinced that it is appropriate, but I would need more information.

Minor issues

1. Title: Consider rewriting this for clarity, e.g. “emotional impact according to the way cancer patients are conducted to the surgical center”. The same applies to the short title

2. Abstract, methods: Consider adding further information, i.e. a sentence on participants’ age (mean/median) and sex, what type of randomisation was carried out, and outlining primary and secondary outcomes.

3. Abstract, Results, lines 41-44: Consider rewriting this - more often satisfied or very satisfied instead of were very satisfied or satisfied?

4. Introduction, line 68: Consider replacing “his” with “their” so the term is not gender specific.

5. Introduction, line 74: Consider replacing “sick situation” with “health condition” or something along these lines.

6. Introduction, line 75: Populations instead of population?

7. Introduction, line 78: Chose instead of choose?

8. Introduction, line 81: “Lack of dignity” instead of “dignity lack”?

9. Introduction, lines 79-83: Consider clarifying this sentence – Wouldn’t those on a stretcher be wearing hospital clothing?

10. Introduction, line 83: “nursing staff” instead of “nurse staff”?

11. Introduction, line 84: consider removing “it is possible to observe”

12. Introduction, line 87: “result in” instead of “reflect on”?

13. Introduction, lines 87-92: Consider rewriting this for clarity. There is a bit of repetition. Also consider replacing “according to our review” with “to our knowledge”. The same applies to the Discussion section, line 273.

14. Results, Table 1: Does Median (Min-Max) refer to age and surgical procedures only? If so, it looks slightly odd to have this above all variables, as most of them are reported using proportions. Consider different ways to show this information. The same applies to Table 3

15. Results, Table 1: Earlier in the manuscript race is mentioned, while table refers to ethnicity. Consider using a single term for consistency. Was ethnicity self-reported?

16. Results, Table 1; 11.7% instead of 11,7% for genitourinary cancers in the stretcher group?

17. Results, footnote for Table 1: Please consider adding the missing data for each variable

18. Results, footnote for Table 1: BMI is defined, but it looks like there are no BMI measures displayed in Table 1?

19. Table 3. first question – Are “doubts” about concerns, uncertainties or both? This is a very broad question, please consider specifying what it means (it may have been an issue with the translation).

20. Table 3. Some of the questions seem irrelevant to the study aims. Was the tool also used to answer different research questions for different studies?

6. PLOS authors have the option to publish the peer review history of their article (what does this mean? ). If published, this will include your full peer review and any attached files.

**Do you want your identity to be public for this peer review?** For information about this choice, including consent withdrawal, please see our Privacy Policy .

Reviewer #1: No

Reviewer #2: No

Reviewer #3: **Yes: ** Octavian Andronic

Reviewer #4: No

---

## [Author Response · Author response to Decision Letter 1]

4 Dec 2024

Journal Requirements:

Answer 1: Thank you very much for your comments. We have reviewed the manuscript and adjusted it to meet the PLOS ONE style requirements, as per your guidance.

Answer 2: Thank you very much for your guidance. I would like to clarify that the submitted protocol does not contain a confidentiality notice, and we agree that it can be shared publicly. The document that contains the institutional logo is the IRB approval letter, and this cannot be removed. We have attached a formal statement signed by the institution's representative, consenting to the publication of the protocol.

Answer 3: Thank you very much for your guidance. We have submitted the database as per your instructions. You can find it through the following link: https://doi.org/10.6084/m9.figshare.27926655.v1

Answer 4: Thank you very much for your guidance. We have included the supporting information as per the journal's instructions.

Reviewers' comments:

Review Comments to the Author

Answer 1: Thank you for your comment above. I would like to clarify that, indeed, some losses occurred during the study. Initially, we screened 239 patients, of whom 45 were found to be ineligible for participation. Randomization was carried out with 194 patients, but 16 were unable to respond to the questionnaires or provide consent, as they entered the operating room before the research team could carry out these procedures. Additionally, there were two refusals to participate, both within the ambulation group. During the process of completing the questionnaires and explaining the consent form, these participants chose not to sign the consent form. In these cases, the questionnaires were discarded in the presence of the participants.

Answer 2: Thank you very much for your suggestion. We have updated the flow diagram and included the requested information.

Answer 3: Thank you very much for your comments. The process followed for data collection was as follows: participants proceeded to the surgical center, where they received a brief explanation about the anxiety, depression, and satisfaction questionnaires before the surgical procedure. After they completed the questionnaires, and once they were finished, the consent process was initiated. This included a detailed explanation of the study and the reason for signing the consent form at a later stage. The Informed Consent Form (ICF) was signed only if the participant agreed to take part in the study; otherwise, the questionnaire was immediately discarded in their presence. I am re-inserting here the text from the manuscript that explains the rationale behind the research team conducting the informed consent process after the intervention and completion of the questionnaires. We do not believe that this procedure introduces selection bias; on the contrary, it ensured that study participants were not influenced into believing that the ambulation intervention might be superior.

“The patient’s study consent was carried out a posteriori, in other words, Informed Consent Form (ICF) application was carried out after the study procedures (conduction and questionnaires’ application) to avoid bias related to the questionnaires’ answers. The data collection process followed these steps: participants were directed to the surgical center, where they received a brief explanation about the anxiety, depression and satisfaction questionnaires and then answered the questions. After completing the questionnaires, the consent process was initiated, with a detailed explanation of the study and the reason why the signature was carried out a posteriori. The Informed Consent Form (ICF) was signed only if the participant agreed to take part in the study; otherwise, the questionnaire was immediately discarded in their presence. The objective was not to induce the patient to believe that being taken to the surgical center walking would be more comfortable and cause greater satisfaction”. This information has been included in the manuscript and can be found in line 128-138.

Answer 4: We appreciate your comment and question. I would like to emphasize that all participants selected for the study met the pre-established exclusion criteria, which included individuals who, shortly before surgery, experienced an adverse event that could impair their ability to walk to the surgical center; patients with physical disabilities or those requiring walking assistance; those indicated for postoperative Intensive Care Unit (ICU) admission; individuals with a prior diagnosis of psychiatric disorders and/or use of anxiolytic and antidepressant medications; and patients who were unaccompanied on the day of surgery. This information can be found in line 93-101.

Answer 5: Thank you for your comments and questions: The theoretical model used was a comparison between groups, and the randomization method employed was simple randomization without matching.

Answer 6: We appreciate a lot your comment. Some items on the checklist are not applicable to the study, such as:

- Item 3b, as there were no changes in the inclusion and exclusion criteria during the study.

- Items 6a and 7b, as no interim analyses were conducted.

- Item 6b, as there were no changes to the study results after the study began.

- Item 14b, as the study was not interrupted.

We have updated and submitted the CONSORT with the other necessary adjustments.

Reviewer #2:

Answer (a): Thank you for your comment. The randomization list was developed by the statistician, and the research team had access to the randomization list only to carry out the random allocation of the patients.

Answer (b): Thank you for your comment. The randomization was generated through the REDCap platform, creating a random list for study entry. The theoretical model used was a comparison between groups, and the randomization method employed was simple randomization without matching. This entire process was designed by the study's statistician. This information has been included in the manuscript and can be found in lines 155-157.

Answer c: Thank you for your comment. The statistical test used for this analysis is the Fisher's Exact Test (2-sided). The effect size was estimated based on Cohen, J. (1988), which provides guidelines for interpreting effect sizes. For this analysis, the desired effect size is approximately 0.43, which is considered a medium effect according to Cohen's benchmarks.

Answer (d): Thank you for your comment. The Mann-Whitney test was used due to the non-Gaussian distribution of the quantitative variables.

Answer (d2): Thank you for your comment. The study did not use statistical modeling for the "group" outcome, as it was not necessary to estimate functional relationships between significant sociodemographic and clinical factors that could explain the predictive difference between groups. Additionally, the observed frequencies at some levels of qualitative variables were insufficient, which could lead to underestimated results and potentially bias the inferences in multivariate models, which require adherence to several assumptions. Furthermore, we do not see the need for this statistical analysis, as the randomization process already equally distributed sociodemographic, clinical, and tumor-related characteristics between the intervention and control groups, as shown in Table 1.

2. Results & Conclusions:

Answer (a): Thank you for your comment. We performed a check and confirmed that all samples with significance are accompanied by the p-value.

Answer (b): Thank you for your comment and suggestion. I would like to emphasize that we have included this suggestion as one of the limitations of our study, as it is not multicentric and includes only a Brazilian population using the public health system. We have modified the highlighted section and added the requested information. This information has been included in the manuscript and can be found in lines 331-335.

Reviewer #3:

Answer: Thank you for your comments. We agree with your comments; however, the study demonstrated that although there was no reduction in anxiety and depression levels, patients who were ambulated to the surgical center in their own clothing reported greater satisfaction regarding the mode of transport (70.5%), the waiting time for surgery (91.5%), and the option to wear their own clothes (100.0%). These findings highlight the potential benefits that this practice may offer to patients with fewer comorbidities.

Reviewer #4:

Major issues

Answer 1: Thank you for your comment. The randomization was generated through the platform REDCap, creating a random list for study entry. The theoretical model used was a comparison between groups, and the randomization method was simple randomization without matching. This entire process was carried out by the study's statistician. This information has been included in the manuscript and can be found in lines 155-157.

Answer 2: Thank you for your comment. We rephrased the sentence as instructed to: "The development of the questionnaire was motivated by the lack of suitable instruments to assess patient satisfaction with healthcare services, as the existing questionnaires were inadequate to meet the specific objectives of the study."This information has been included in the manuscript and can be found in lines 121-124.

Answer 3: Thank you for your comment. This study did not have blinding criteria, meaning that both the patients and the researchers were aware of the intervention used (stretcher transport or ambulation). We would like to clarify that the three outcomes—anxiety, depression, and satisfaction—were considered primary outcomes (primary objective). However, we emphasize that for the sample size calculation, we referred to the study by Kojima et al. (7), which assessed anxiety levels.

Answer 4: Thank you for your comment. The data collection process was as follows: participants proceeded to the surgical center, where they received a brief explanation about the anxiety, depression, and satisfaction questionnaires prior to the surgical procedure. They then completed the questionnaires, and after finishing them, the informed consent process was initiated, with a detailed explanation of the study and the reason for signing the consent form at a later time. The Informed Consent Form (ICF) was signed only if the participant agreed to take part in the study; otherwise, the questionnaire was immediately discarded in their presence. We do not believe that this procedure introduces selection bias. On the contrary, this process ensured that study participants were not influenced into thinking that the ambulation intervention might be superior. This information has been included in the manuscript and can be found in lines 130-136.

Answer 5: Thank you for your comment. The statistical test used was the Fisher's Exact Test (2-sided). The effect size was estimated to be approximately 0.43, based on Cohen, J. (1988), and the proportions considered from the Kojima et al. (7) study refer to the "Calm/mildly anxious" group.

Answer 6: Thank you for your comment. We agree that the results of our study may apply to a specific population: cancer patients who are Brazilian, treated at a reference cancer center, and users of the public healthcare system. Since our institution serves 60% of patients from the state of São Paulo, we believe this explains the high proportion of white patients (over 50%). We do not believe this fact limits the generalizability of our findings, as we employed an appropriate design to address the research question. We agree that ideally, this same study should be conducted in other populations, from different countries and cultural background. We emphasize that we have included the above-mentioned aspects as a potential limitation of our study. This information has been included in the manuscript and can be found in lines 332-338.

Answer 7: Thank you for your comment and suggestion, very well detected. We have revised the text in the abstract to ensure greater consistency of the information: “The transportation method to the surgical center did not affect anxiety and depression levels in cancer patients with no or low fall risk. However, patients in the ambulation group were more satisfied with the waiting time for surgery and the type of clothing”. This information has been included in the manuscript and can be found in lines 44-46.

Answer 8: Thank you for your comment. The recruitment was carried out consecutively. The difference between the groups occurred due to logistical difficulties related to hospital routines and coordination between the ward and surgical center teams, which resulted in 30 participants not following the randomization. Of these, 12 patients were randomized to be transported to the surgical center by stretcher but were instead transported by walking, and 18 patients who had been randomized to walk to the surgical center were transported by stretcher. Since the aim of the study was to assess satisfaction, anxiety, and depression levels according to the type of transportation to the surgical center, we decided to maintain the analysis of these patients based on the actual mode of transport they experienced. We emphasize that we have revised our flowchart to make the recruitment and randomization process clearer, including refusals, losses, and the corresponding reasons. Below is the updated flowchart.

Answer 9: Thank you for your comment. The waiting time for the two modes of transport was similar. The difference lies in the location where the patient waits prior to the surgery. Patients transported on stretchers wait in the ward room, lying in bed and wearing hospital gowns, which reinforces the patient's ill condition. In contrast, patients who walk to the surgical center wait in the reception area of the ward, dressed in their own clothes. We believe that waiting in the company of others in similar situations, along with the exchange of experiences, resulted in greater satisfaction regarding the waiting time for the surgical procedure. This information is included in the manuscript, and can be found on lines 266-270.

Answer 10: Thank you for your comment. We do not believe that the mode of transportation to the surgical center influenced the patients' preference for the mode of transport, as both groups were asked at the end of the satisfaction questionnaire whether they preferred to be transported by stretcher, wheelchair, or walking. This question was asked before the consent was signed and before entering the study. Therefore, at the time of the question, the patients were unaware that they could be part of a study evaluating the mode of transportation. It is important to emphasize that the stretcher group was not informed that walking to the surgical center was an option or that another group was being transported in this manner.

Answer 11: Thank you for your comment. During the administration of the questionnaire, the research team clarified that satisfaction was not related to the surgery scheduling time, but rather to the waiting time from arrival at the hospital until transfer to the surgical center.

Answer 12: Thank you for your comment. The inclusion of the question regarding satisfaction with clothing and the participants' preferences was a suggestion from the research team after the start of the study and patient recruitment, which is why there are missing data regarding this information.

Answer 13: Thank you for your comment. We would like to clarify that the three outcomes—anxiety, depression, and satisfaction—were considered primary outcomes (primary objective). However, we emphasize that for the sample size calculation, we referred to the study by Kojima et al. (7), which assessed anxiety levels. We reported to our IRB the patients who did not adhere to the randomization protocol as protocol deviations. However, since we conducted the analyses of anxiety, depression, and satisfaction in the same manner, respecting the mode of transport the patient was assigned to, we do not believe this affects the study's results.

Answer 14: Thank you for your comment. Firstly, we believe that if the patients transported on stretchers had been wearing their own clothes, it would not have inf

---

## [Decision Letter · Decision Letter 1]

3 Jan 2025

PONE-D-24-33555R1Emotional impact according to the way cancer patients are conducted to the surgical center: a randomized clinical trial comparing ambulation to the stretcherPLOS ONE

Dear Dr. Oliveira,

Thank you for submitting your manuscript to PLOS ONE. We are pleased to inform you that the reviewers find your revised manuscript suitable for publication, pending one minor revision suggested by a reviewer.

We look forward to receiving your revised manuscript.

Kind regards,

Alexandre Bonatto

Academic Editor

PLOS ONE

Journal Requirements:

Reviewers' comments:

Reviewer's Responses to Questions

**Comments to the Author**

1. If the authors have adequately addressed your comments raised in a previous round of review and you feel that this manuscript is now acceptable for publication, you may indicate that here to bypass the “Comments to the Author” section, enter your conflict of interest statement in the “Confidential to Editor” section, and submit your "Accept" recommendation.

Reviewer #1: All comments have been addressed

Reviewer #2: All comments have been addressed

Reviewer #4: (No Response)

2. Is the manuscript technically sound, and do the data support the conclusions?

Reviewer #1: Yes

Reviewer #2: (No Response)

Reviewer #4: Yes

3. Has the statistical analysis been performed appropriately and rigorously?

Reviewer #1: Yes

Reviewer #2: (No Response)

Reviewer #4: I Don't Know

4. Have the authors made all data underlying the findings in their manuscript fully available?

Reviewer #1: Yes

Reviewer #2: (No Response)

Reviewer #4: Yes

5. Is the manuscript presented in an intelligible fashion and written in standard English?

Reviewer #1: Yes

Reviewer #2: (No Response)

Reviewer #4: Yes

6. Review Comments to the Author

Reviewer #1: (No Response)

Reviewer #2: (No Response)

Reviewer #4: The authors have addressed my queries thoroughly and carefully. I have reviewed their responses and the amended version with tracked changes. I only have a few minor comments the authors may wish to consider:

1. The amended flowchart looks better, but the box “Did not follow randomisation” is still confusing as the numbers underneath do not make much sense (i.e. numbers do not add up). If those not following randomisation still took part, what about having both the numbers for those following and those not following randomisation? Similarly, this could also be specified in the final two boxes, both for the stretcher group and the ambulation group (this information was provided in the response to reviewers, but I could not find it in the amended figure). Consider using the CONSORT template as it accounts for these issues: https://pmc.ncbi.nlm.nih.gov/articles/PMC2844943/

2. Text under “This is the Figure 1 legend” (line 173, page 7): eligible instead of elegible? Please note that the figure may also need to be edited.

3. Legends under all tables: what is meant by “ignored values”? Is this the same as missing data?

4. Discussion, page 15, line 336: instead of “with all types of surgical procedures” consider saying “undergoing all types of surgical procedures” or something along these lines

5. Discussion, page 15, line 347: Based on the study findings only, or based on the study findings and other evidence from the literature?

7. PLOS authors have the option to publish the peer review history of their article (what does this mean? ). If published, this will include your full peer review and any attached files.

**Do you want your identity to be public for this peer review?** For information about this choice, including consent withdrawal, please see our Privacy Policy .

Reviewer #1: No

Reviewer #2: No

Reviewer #4: **Yes: ** Natalia Calanzani

---

## [Author Response · Author response to Decision Letter 2]

27 Jan 2025

Answer 1: We appreciate your comment and the detailed considerations. We have updated the flowchart to make it clearer and more informative. We changed the description from 'did not follow randomization' to 'crossover,' indicating that participants switched groups after randomization. Additionally, we included a caption in the flowchart explaining that this group change occurred due to unforeseen logistical issues related to hospital organization. Since the primary objective of the study was to assess anxiety levels based on the mode of transport, we decided to keep these patients in the study, even though they did not follow the initially assigned mode of transport. These changes can be found on page 7, lines 181-184, and in the document titled 'Fig-1,' attached to the submission platform.

Answer 2: Thank you very much for your observation. We have made the correction to the word as pointed out. This change can be found on page 7, line 170, and in the document titled 'Fig-1,' attached to the submission platform.

Answer 3: We appreciate your comment and question. 'Ignored values' refers to missing data. To avoid any possible confusion, we have changed all the captions to 'missing data.' These changes can be found in lines: 188, 217, and 236.

Answer 4: Thank you for your comments and suggestion. We have made the change as suggested. This change can be found in lines 336-337.

Answer 5: Thank you for your comments and suggestion. We have modified the sentence to make it clearer as 'Based on the study findings and other evidence from the literature.' This information can be found in lines 346-347.

---

## [Decision Letter · Decision Letter 2]

26 Feb 2025

Emotional impact according to the way cancer patients are conducted to the surgical center: a randomized clinical trial comparing ambulation to the stretcher

PONE-D-24-33555R2

Dear Dr. Oliveira,

We’re pleased to inform you that your manuscript has been judged scientifically suitable for publication and will be formally accepted for publication once it meets all outstanding technical requirements.

Kind regards,

Karam R. Motawea, MBBCh

Academic Editor

PLOS ONE

Additional Editor Comments (optional):

The authors have addressed the comments of the reviewers adequately. Therefore, I am pleased to inform you that your manuscript has been accepted for publication in PLoS ONE.

Reviewers' comments:

Reviewer's Responses to Questions

**Comments to the Author**

1. If the authors have adequately addressed your comments raised in a previous round of review and you feel that this manuscript is now acceptable for publication, you may indicate that here to bypass the “Comments to the Author” section, enter your conflict of interest statement in the “Confidential to Editor” section, and submit your "Accept" recommendation.

Reviewer #1: All comments have been addressed

Reviewer #2: All comments have been addressed

Reviewer #4: All comments have been addressed

2. Is the manuscript technically sound, and do the data support the conclusions?

Reviewer #1: Yes

Reviewer #2: (No Response)

Reviewer #4: Yes

3. Has the statistical analysis been performed appropriately and rigorously?

Reviewer #1: Yes

Reviewer #2: (No Response)

Reviewer #4: N/A

4. Have the authors made all data underlying the findings in their manuscript fully available?

Reviewer #1: Yes

Reviewer #2: (No Response)

Reviewer #4: Yes

5. Is the manuscript presented in an intelligible fashion and written in standard English?

Reviewer #1: Yes

Reviewer #2: (No Response)

Reviewer #4: Yes

6. Review Comments to the Author

Reviewer #1: All requested adjustments were made appropriately. The article describes important results and will contribute to evidence-based clinical practice.

Reviewer #2: (No Response)

Reviewer #4: (No Response)

7. PLOS authors have the option to publish the peer review history of their article (what does this mean? ). If published, this will include your full peer review and any attached files.

**Do you want your identity to be public for this peer review?** For information about this choice, including consent withdrawal, please see our Privacy Policy .

Reviewer #1: No

Reviewer #2: No

Reviewer #4: **Yes: ** Natalia Calanzani

---

## [Editor Report · Acceptance letter]

PONE-D-24-33555R2

PLOS ONE

Dear Dr. Oliveira,

I'm pleased to inform you that your manuscript has been deemed suitable for publication in PLOS ONE. Congratulations! Your manuscript is now being handed over to our production team.

Kind regards,

on behalf of

Dr. Karam R. Motawea

Academic Editor

PLOS ONE